# Comparative Analysis of Optoelectrical Performance in Laser Lift-Off Process for GaN-Based Green Micro-LED Arrays

**DOI:** 10.3390/nano13152213

**Published:** 2023-07-30

**Authors:** Chuanbiao Liu, Feng Feng, Zhaojun Liu

**Affiliations:** 1Harbin Institute of Technology, Harbin 150006, China; 11949036@mail.sustech.edu.cn; 2Department of Electrical and Electronic Engineering, The Southern University of Science and Technology, Shenzhen 518000, China; 3Department of Electronic and Computer Engineering, Hong Kong University of Science and Technology, Hong Kong SAR 999077, China; ffengaa@connect.ust.hk

**Keywords:** GaN-based Micro-LEDs, optoelectrical performance, laser lift-off, Micro-LED display

## Abstract

This work explores the pivotal role of laser lift-off (LLO) as a vital production process in facilitating the integration of Micro-LEDs into display modules. We specifically investigate the LLO process applied to high-performance gallium nitride (GaN)-based green Micro-LED arrays, featuring a pixel size of 20 × 38 μm on a patterned sapphire substrate (PSS). Scanning electron microscopy (SEM) observations demonstrate the preservation of the GaN film and sapphire substrate, with no discernible damage. We conduct a comprehensive analysis of the optoelectrical properties of the Micro-LEDs both before and after the LLO process, revealing significant enhancements in light output power (LOP) and external quantum efficiency (EQE). These improvements are attributed to more effective light extraction from the remaining patterns on the GaN backside surface. Furthermore, we examine the electroluminescence spectra of the Micro-LEDs under varying current conditions, revealing a slight change in peak wavelength and an approximate 10% decrease in the full width at half maximum (FWHM), indicating improved color purity. The current–voltage (I–V) curves obtained demonstrate the unchanged forward voltage at 2.17 V after the LLO process. Our findings emphasize the efficacy of LLO in optimizing the performance and color quality of Micro-LEDs, showcasing their potential for seamless integration into advanced display technologies.

## 1. Introduction

Micro-LED display technology have received great attention recently [1]. Numerous companies and researchers worldwide have conducted extensive studies on Micro-LED displays since 2000 [2,3,4,5,6,7]. With China as a frontrunner in electronics development, its Micro-LED industry is poised for rapid growth in the coming years, driven by the escalating demand for various applications such as TVs, indoor large-screen displays, small and medium-sized laptop displays, tablet displays, portable devices, heads-up displays (HUDs), head-mount displays (HMDs), augmented reality (AR), virtual reality (VR), and other terminal applications [8,9,10]. The Micro-LED display technology represents a major trend in next-generation semiconductor display technology, enabling remarkable advantages, including ultralow power consumption, ultrahigh current and luminous density, ultrasmall size, ultrahigh resolution, and fast response speed [11,12,13,14,15].

Sapphire substrate has emerged as the preferred choice for epitaxial growth of GaN materials in Micro-LED chips due to its low lattice mismatch with GaN and cost-effectiveness. However, the nonconductive nature and poor thermal conductivity of sapphire substrate have adverse effects on the luminous efficiency of Micro-LED devices. Additionally, the brittleness of sapphire hampers the application of Micro-LEDs in flexible and ultrathin displays [16,17,18]. Consequently, the separation of the sapphire substrate becomes a critical and essential process, where lift-off technology emerges as a viable solution. Among various lift-off techniques such as chemical lift-off, mechanical lift-off, and ion lift-off, laser lift-off (LLO) technology offers several advantages, including high energy input efficiency, minimal device damage, equipment flexibility, and versatile application modes. As a result, LLO has become a pivotal technology for manufacturing flexible electronic devices [19,20,21,22,23].

In the LLO process, ultraviolet lasers are employed as the photon energy of this light source surpasses the bandgap of GaN but remains below that of sapphire. This selective absorption of laser energy by the GaN layer causes a rapid increase in temperature within the interface region. The localized heating reaches the thermal delamination temperature, leading to the thermal decomposition of GaN into nitrogen gas (N_2_) and low-melting-point gallium metal (Ga). This process occurs within the temperature range of 900 °C to 1000 °C, facilitating the separation of the GaN epilayer from the sapphire substrate [24,25].

In this paper, we undertake the design and fabrication of Micro-LED arrays on sapphire substrates. The main focus of our study is the separation of GaN from the patterned sapphire substrate (PSS) using the LLO technique. Additionally, we conduct a thorough comparison and analysis of the photoelectric characteristics before and after LLO. Specifically, we measure and discuss the light output power (LOP) and external quantum efficiency (EQE) for individual Micro-LED pixels. We also examine and study the difference in peak wavelength, full width at half maximum (FWHM), and blue shift before and after LLO. To complement our study, scanning electron microscopy (SEM) was employed to observe the morphology of the Micro-LED arrays and the cross-section morphology of GaN and sapphire.

## 2. Materials and Methods

In this experiment, we utilized a 4 inch PSS substrate epitaxial, where the Micro-LED array structure consists of multiple layers from bottom to top: sapphire substrate layer, GaN buffer layer, N-type GaN layer, active layer with multiple quantum wells (MQWs), P-type GaN contact layer, current spreading layer (CSL) and p-type electrode. The combined thickness of the epitaxial layers is approximately 6 μm, while the sapphire substrate thickness measures approximately 450 μm.

The mesa structure was prepared by inductively coupled plasma (ICP) etching GaN down to the n-GaN layer to facilitate the ohmic contact process of the n electrode. Subsequently, a thick layer of SiO_2_ protection was deposited using plasma-enhanced chemical vapor deposition (PECVD), and then wet etching was employed to remove SiO_2_ around the chip area. The following step involved using ICP to etch GaN onto sapphire substrate, forming independent pixel units. Afterward, the p and n electrode layers were created by electron beam evaporation, and a SiO_2_ passivation layer was deposited via PECVD. The SiO_2_ in the n and p electrode regions was etched by ICP. Finally, the contact electrodes of p and n were made using electron beam evaporation, followed by ultrasonic stripping. 

Each individual green Micro-LED chip had a size of 20 × 38 μm, with a pixel horizontal distance of 6 μm and vertical distance of 50 μm between two adjacent chips. Figure 1a depicts the schematic diagram of Micro-LED, while Figure 1b shows the scanning electron microscope (SEM) morphology of Micro-LED arrays.

After preparing the Micro-LED arrays, we proceeded to characterize their optoelectrical performance. The current-voltage (I-V) characteristics were measured using a Keysight B1500 Analyzer. Additionally, we obtained the electroluminescence (EL) spectra, along with radiometric power measurements, using an Ocean Optics USB 2000+ spectrometer with a CC-3 cosine corrector.

Once the optoelectrical performance test was completed, the subsequent LLO was performed to remove the sapphire substrate. The principle behind LLO involves using short-wavelength laser with photon energy greater than the GaN bandgap but smaller than that of sapphire to irradiate one side of sapphire. The laser is strongly absorbed by the surface GaN after passing through sapphire [26]. In this experiment, the photon energy of the all-solid-state semiconductor laser used was 4.83 eV, falling between the bandgaps of sapphire substrate (E_S_: 9.9 eV) and GaN (E_GaN_: 3.4 eV), as shown in Figure 2. The laser absorption occurred at the GaN–sapphire interface, leading to decomposition of undoped GaN into nitrogen (N_2_) and gallium (Ga) metals.

In this experiment, a state-of-the-art ultrafast pulse laser was used. The chemical reaction occurring at the interface can be expressed as follows [27]:2GaN (solid) → 2Ga (liquid) + N_2_ (gas).(1)

Figure 3 illustrates the LLO process. As shown in Figure 3b, the green Micro-LED array with the sapphire side facing up was positioned on the laser experimental platform. As seen in Figure 3c,d, a 257 nm solid-state laser was employed to scan and lift off the sapphire surface. During the LLO process, GaN underwent decomposition into N_2_ and metal Ga, successfully separating from the sapphire substrate. As shown in Figure 3e, a new blue tape was applied to cover the stripped Micro-LED array, ensuring a more cohesive arrangement of the Micro-LED chips. Subsequently, as shown in Figure 3f, the original blue tape was removed.

## 3. Results and Discussion

### 3.1. LLO Results and Discussion

Several factors impact the performance of Micro-LEDs due to the quality of the LLO process. These factors include delayed N_2_ pressure, Ga agglomeration, and electrode shedding [28,29,30]. Following extensive and thorough verification and comparison, we opted to directly adopt the following mature LLO parameters: wavelength, 257 nm; power, 0.8 W; fill density, 13 μm; pitch, 13 μm; frequency, 200 kHz; scan speed, 2600 mm/s.

After conducting the LLO, we observed that there was no chip fragmentation or chip electrode falling off. The all-solid-state semiconductor laser with a wavelength 257 nm proved to be a highly stable laser source, ensuring a smooth peeling interface with no accumulation of metal residues. SEM images from different dimensions further confirmed the success of the LLO process. To demonstrate our accurate control of LLO parameters, we selected a region without LLO, shown in Figure 4a, where the boundary was clearly defined, and the separation was complete. Figure 4b exhibits the thoroughness of the LLO process and the successful transfer. The smoothness and integrity of the PSS interface are demonstrated in Figure 4c, with no evident epitaxial layer residue at the sapphire interface after laser stripping. The circle of the SiO_2_ insulating layer around the Micro-LED chip remained undamaged by the laser. In Figure 4d, no visible damage can be observed between the GaN film and the sapphire substrate, indicating the complete removal of the GaN film after LLO. The UV laser’s highly concentrated light focused on the n-GaN/sapphire surface, providing sufficient power for the semiconductor to turn into gas. Optimized nitrogen venting during the process, combined with line-by-line laser scanning, resulted in a smooth lift-off surface.

### 3.2. Optical Characterization 

Light output power (LOP) and the external quantum efficiency (EQE) values of Micro-LED arrays before and after LLO were calculated using the following equation [31]:(2)$$EQE=\frac{e\cdot P}{E\cdot I},$$
where E represents the averaged photon energy across the emission peak, while e, P, and I correspond to the electron charge, radiometric power, and injection current, respectively. The LOP of different Micro-LEDs was integrated using EL spectra, which were measured under the same conditions for comparative analysis.

The patterned prism structures on the backside of the lifted off Micro-LED device remain almost the same, contributing significantly to improve light extraction compared to the device before the LLO process [32]. Furthermore, the removal of the sapphire substrate reduces the light absorption, allowing more light rays to be extracted from Micro-LED. As a result, Figure 5a,b show that the LOP and EQE for a single device after the LLO process were improved by 18% and 21%, respectively, in comparison to a Micro-LED on substrate. The EQE reached its peak at a higher injection current (J_peak_) for lifted-off devices. This notable EQE improvement can also be attributed to the enhanced thermal dissipation of the device structure, enabling the J_peak_ to reach a higher value [33].

Two probes with a diameter of 0.5 μm were carefully inserted onto the n pad and the p pad, which were less than 15 μm wide. With higher current injection, the green Micro-LED device exhibited a more pronounced light output, as shown in Figure 6. The electroluminescence spectrum (EL) with different current before LLO is displayed in Figure 7a, while the spectrum after LLO is presented in Figure 7b, encompassing the range of 300–800 nm.

Following the LLO process, the Micro-LED exhibited a slight blue shift in the peak wavelength. This shift can be attributed to the partial release of stress in the epitaxial layer, resulting in a weakening of the semiconductor band quantum-confined Stark effect (QCSE), consequently increasing the bandgap and inducing the blue shift in wavelength. Upon comparing Figure 7a,b, it is evident that the spectral radiation intensity after LLO significantly surpassed that before LLO. This increase can be attributed to the reduction in dislocation density after LLO, and the improvement in the light extraction rate due to the removal of sapphire.

Figure 7c illustrates the peak wavelength of the green Micro-LED, showing a noticeable blue shift in the peak wavelength as the current increases. Specifically, before the laser lift-off (LLO) process, the peak wavelength shifted from 528 nm at 10 μA to 522 nm at 100 μA. Similarly, after LLO, the peak wavelength shifted from 529 nm at 10 μA to 523 nm at 100 μA. This blue shift can be attributed to the influence of the QCSE, which is further influenced by the screen effect [34]. The blue shift was not pronounced, possibly due to the epitaxial layer having better lattice quality and a less spontaneous polarization effect.

The FWHM is shown in Figure 7c, with mean values before and after LLO of 33 nm and 30 nm, respectively. The strain changes caused the InGaN bandgap to tilt in MQWs, leading to peak shift. This phenomenon can be attributed to the QCSE [35], caused by the shrinking bandgap between the valence band and conduction band, as well as changes in the electric field. Thermal radiation causes changes in indium spreading in MQWs, with the laser providing heat in the LLO process.

In this experiment, the FWHM did not perform optimally, possibly due to several factors influencing it. However, the main reason could be attributed to the natural inhomogeneity in the distribution of In in the InGaN quantum well layer. The FWHM improved after LLO, probably due to the influence of high temperature in the LLO process, which resulted in a more uniform redistribution of In in the InGaN quantum well, leading to a smaller gap in the back half height of LLO.

### 3.3. Electrical Characterization and Uniformity

The I-V characteristics and the semi-log plot of the I-V relation are displayed in Figure 8a, and data analysis was performed on nine different periodically selected positions. The forward voltage was measured at 2.17 V with a current of 1 μA before LLO, and the leakage current was found to be less than 1 nA. After LLO, the forward voltage remained the same, indicating no significant difference in the I-V characteristics of the Micro-LED chip before and after LLO. The LLO process had a minimal effect on the optoelectrical performance of the Micro-LED chips, with highly uniform I-V curves observed in all selected devices. The leakage current at a reversed voltage of −3 V was smaller than 1 nA, demonstrating excellent rectified characteristics.

The EL measurements provided evidence that the Micro-LED after LLO exhibited comparable quality and stability to the original devices. The distribution maps in Figure 8b illustrated the normalized EL intensity (at V_F_ = 5 V) of periodically selected pixels. The root mean square (rms) and standard deviation (σ) values were determined as 0.695 and 0.06292, respectively. Consequently, the devices in the left part of the array conducted a higher current, resulting in a higher EL intensity. This intensity disparity was tentatively attributed to the uneven growth of the Al-GaN, multiple quantum well (MQW), and GaN layers during metal–organic chemical vapor deposition (MOCVD).

## 4. Conclusions

In conclusion, our study focused on characterizing the electrical properties of green Micro-LED arrays measuring 20 × 38 μm. The main investigation revolved around the laser lift-off (LLO) process, which demonstrated no significant damage between the gallium nitride (GaN) film and the sapphire substrate. The patterned structures on the backside of the Micro-LEDs remained intact after removing the patterned sapphire substrate (PSS) during LLO. Comparing the optoelectrical performance before and after LLO, we observed notable improvements. The Micro-LED devices exhibited enhanced light output power (LOP) and external quantum efficiency (EQE) due to improved light extraction and heat dissipation. We also noted changes in the peak wavelength and full width at half maximum (FWHM) of the Micro-LEDs, with the FWHM reduced by approximately 10%. However, the forward voltage remained nearly unchanged before and after LLO, measuring around 2.17 V according to the current-voltage (I–V) curve analysis. This suggests that the LLO process had a minimal effect on the electrical characteristics of the Micro-LEDs.

## Figures and Tables

**Figure 1 nanomaterials-13-02213-f001:**
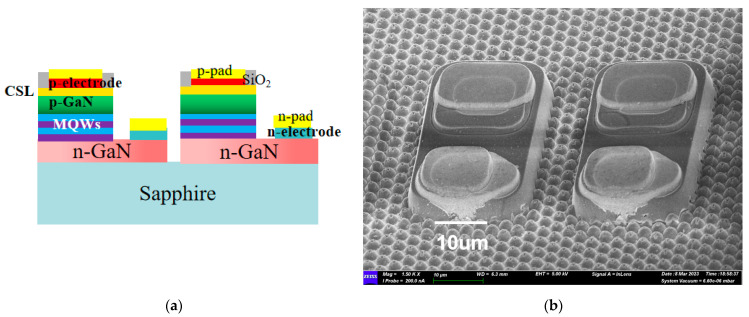
(**a**) The schematic diagram of Micro-LED; (**b**) fabricated Micro-LED arrays using scanning electron microscope morphology.

**Figure 2 nanomaterials-13-02213-f002:**
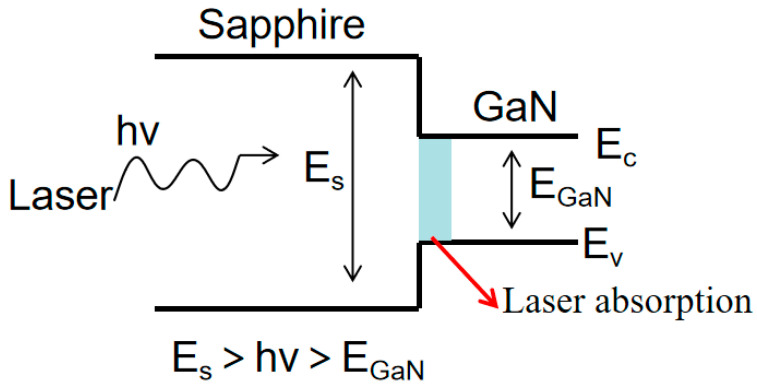
Schematic diagram of physical mechanism of LLO process.

**Figure 3 nanomaterials-13-02213-f003:**
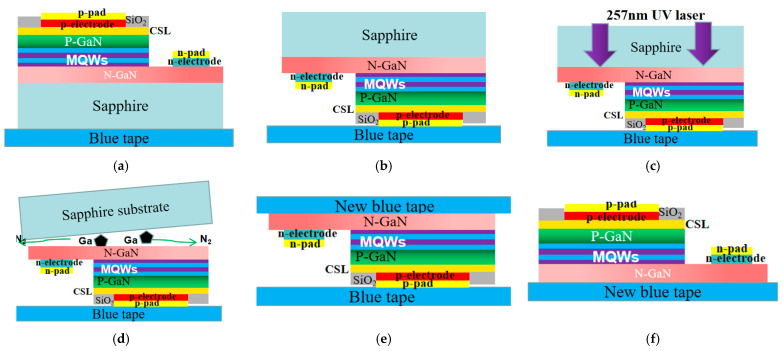
LLO mechanism and process: (**a**) original Micro-LED arrays; (**b**) sapphire substrate facing up; (**c**) LLO process; (**d**) GaN and sapphire separation; (**e**) the new blue tape is glued onto the Micro-LED array; (**f**) the original blue tape is removed.

**Figure 4 nanomaterials-13-02213-f004:**
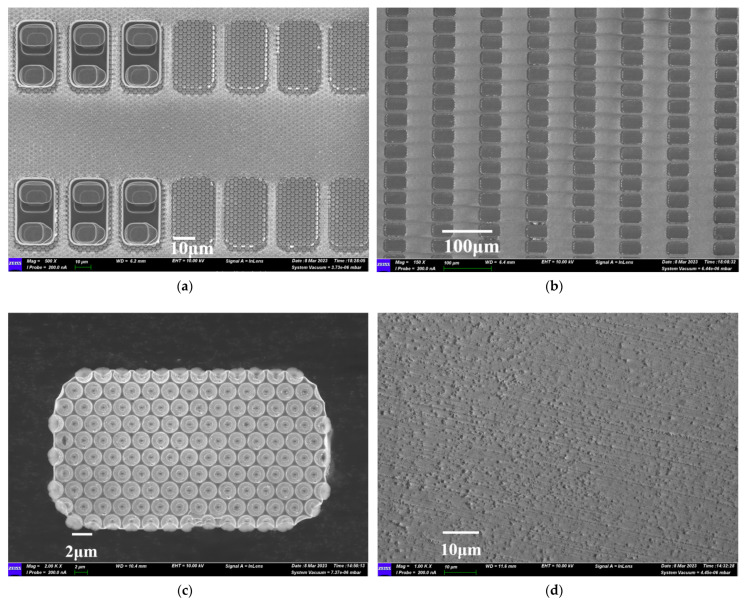
Scanning electron microscope morphology after LLO: (**a**) the region of LLO and without LLO; (**b**) the sapphire substrate after LLO; (**c**) bottom of single Micro-LED after LLO; (**d**) the areas without Micro-LED after LLO.

**Figure 5 nanomaterials-13-02213-f005:**
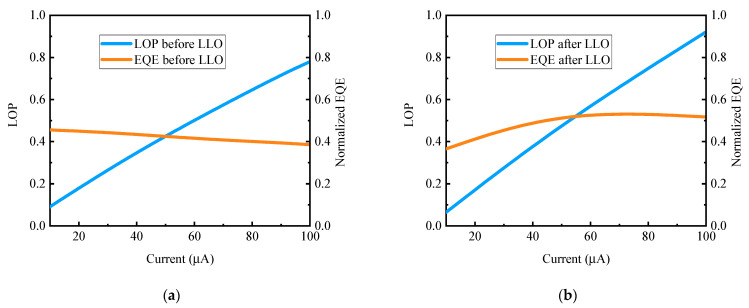
(**a**) LOP and EQE variation with current before LLO; (**b**) LOP and EQE variation with current after LLO.

**Figure 6 nanomaterials-13-02213-f006:**
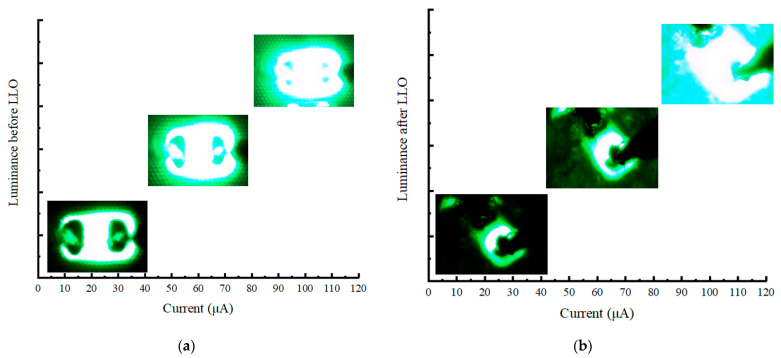
(**a**) Luminance photos before LLO at 10 μA, 50 μA and 100 μA, respectively; (**b**) luminance photos after LLO at 10 μA, 50 μA, and 100 μA.

**Figure 7 nanomaterials-13-02213-f007:**
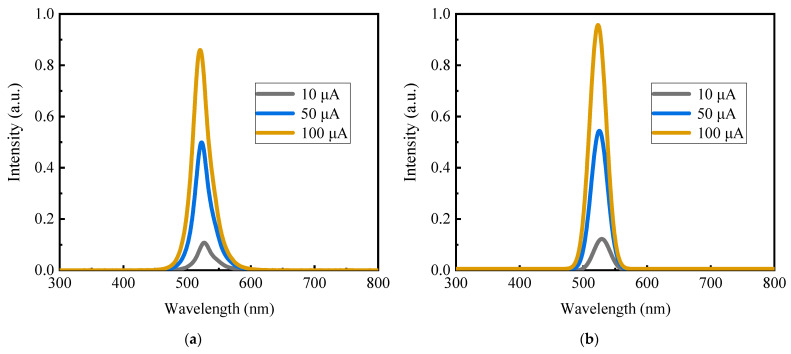
(**a**) Electroluminescence spectrum with different current before LLO; (**b**) electroluminescence spectrum with different current after LLO; (**c**) the peak wavelength and FWHM variation with current.

**Figure 8 nanomaterials-13-02213-f008:**
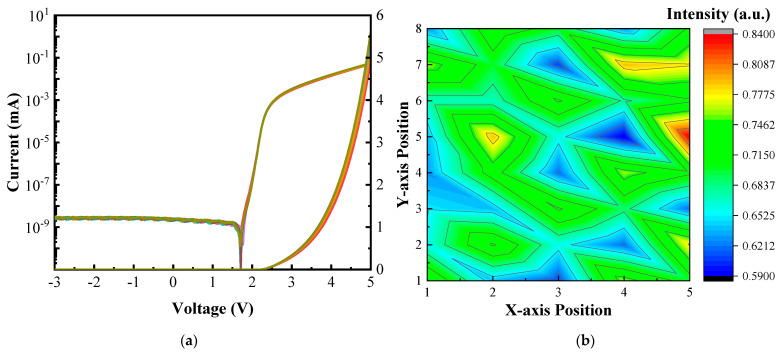
(**a**) Linear I-V with semi-log I-V plot from −3 to 5 V and I-V characteristic uniformity; (**b**) EL intensity distribution uniformity.

## Data Availability

Not applicable.

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
