# Peer review of "Comparative Analysis of Optoelectrical Performance in Laser Lift-Off Process for GaN-Based Green Micro-LED Arrays"

_nanomaterials, 2023, doi:10.3390/nano13152213_

Round 1
Reviewer 1 Report
The manuscript presents information about the Comparative Analysis of Optoelectrical Performance in Laser Lift-Off Process for GaN-based Green Micro-LED Arrays.
This article investigates the LLO process of high-performance gallium nitride (GaN) based green Micro-LED arrays with a pixel size of 20 × 38 μm on a patterned sapphire substrate. Morphology, optical, electroluminescence characterization techniques were used for experimental investigations.
However, in the current form, this paper is not suitable for publication. It certainly needs a major revision for the following reasons:
There is no analysis of obtained results. Just only representation of experimental data. The comparative study with literature data should be represented. The readability of the manuscript should be improved significantly.
Practically no information about the LLO process. How were estimated optimal modes for removing the sapphire substrate?
It is not clearly how was defined the mesa structure of each pixel using photolithography and inductively coupled plasma (ICP)?
The sentence "The energy of the light source was adequately controlled, the focusing effect was good, the peeling interface was smooth, and there was no metal residue of accumulation".... should be quantitatively estimated. It is just general words.
Fig. 6 a, b demonstrate the EL spectrum. The Y-axis should be denoted as Intensity instead of Spectral Irradiance.
Traditionally, the Light output power (LOP) and the external quantum efficiency (EQE) of light sources estimate using integrating sphere.
The PQLY should be analyzed with standard sample for comparative study.
For the equation (2) the authors should represent the references where such approach used for LOP and EQE calculation.
In this case the standard deviation should be represented in Fig. 4 a, b
It is not clear how the authors received the value of LOP and EQE at 18 and 21%. In all Fig. they demonstrated Normalized values.
Did the author estimate the semiconductor layers of investigated structures?
Minor editing of English language required
Author Response
Thanks for your review,Please see the attachment .

Reviewer 2 Report
This is a very good and interesting paper dealing with laser lift-off technique for micro-LEDs. The paper is very well written, clear and the experimental results are sound. The paper has excellent archival potential. I suggest to accept the paper provide that authors take in to account a couple of almost "decorative" recommendations:
1. Please add an abbreviations glossary ate the beginning of the paper. Some acronyms (e.g.,PSS are still not explicit…).
2.Please give the luminance values in cd/m2 in figure 5 and add some comments about these values in the text.
3. Please use the greek character "µ" instead of "u" when you express micro-amperes etc. (e.g., µA than uA…)
4. What representing the various coloured lines in figure 7a. Please add a comment or just keep a single curve.
Author Response
Thank you for your review,the response,Please see the attachment.

Round 2
Reviewer 1 Report
Authors changed the manuscript in accordance with the comments. Accept
Minor editing of English language required
